# Research on Performance Optimization of Novel Sector-Shape All-Vanadium Flow Battery

**Kai Sun, Mengyao Qi, Xinrong Guo, Weijia Wang, Yanqiang Kong, Lei Chen *, Lijun Yang and Xiaoze Du**

Key Laboratory of Power Station Energy Transfer Conversion and System of Ministry of Education, School of Energy Power and Mechanical Engineering, North China Electric Power University, Beijing 102208, China; 120222202092@ncepu.edu.cn (K.S.); 15128164563@163.com (M.Q.); 120222202451@ncepu.edu.cn (X.G.); wangwj@ncepu.edu.cn (W.W.); kongyq@ncepu.edu.cn (Y.K.); yanglj@ncepu.edu.cn (L.Y.); duxz@ncepu.edu.cn (X.D.)
* Correspondence: leichen@ncepu.edu.cn

**Abstract:** The all-vanadium flow batteries have gained widespread use in the field of energy storage due to their long lifespan, high efficiency, and safety features. However, in order to further advance their application, it is crucial to uncover the internal energy and mass transfer mechanisms. Therefore, this paper aims to explore the performance optimization of all-vanadium flow batteries through numerical simulations. A mathematical and physical model, which couples electrochemical reactions and thermal mass transfer processes within a novel sector-shape all-vanadium flow battery, has been established. Subsequently, the impact of cell thickness and operating parameters on the distribution of various physical fields and performance parameters has been investigated. The results show that the potential and overpotential decrease as the electrode thickness increases, while the energy efficiency initially rises and then declines. As for operating parameters, higher electrolyte concentration demonstrates superior performance, while changes in electrolyte flow and current density have comprehensive effects on the battery. The cell performance can be adjusted based on the integrated mass transfer process and energy efficiency.

**Keywords:** all-vanadium flow battery cell; structural optimization; electrochemical performance; electrochemical energy storage; numerical simulation; internal energy and mass transfer; sector-shape; energy efficiency



## 1. Introduction

In the current era of significant transformation in the energy industry, achieving peak carbon emissions by 2030 and carbon neutrality by 2060 serves as a strategic guideline for Chinese development and a global goal for the advancement of green energy. To achieve the so-called dual carbon goals, the development of energy storage technology is crucial. Among the various energy storage methods available, electrochemical energy storage holds immense potential with its wide range of applications. For instance, it can be utilized for grid peak shaving, valley filling, and ensuring stable and secure grid connections for renewable energy sources. The development of electrochemical energy storage is a key pathway toward achieving the "double-carbon" objective [1]. As one of the most studied flow batteries, the all-vanadium flow battery (VFB) stands out due to its advantages in large-scale energy storage, such as site flexibility, high efficiency, and long lifespan. Compared to other novel flow batteries, it also shows high power and more robust chemistry. For zinc–iron redox flow battery (ZIRFB), there are always zinc dendrites limiting areal capacity on the anode, which may lead to the short-circuit and reduction in the battery lifetime [2]; the nonaqueous organic redox flow battery (NAORFB) faces the key challenges such as high electroactive material cost and low energy density [3]. Notably, the VFB also boasts excellent safety and environmental performance [4]. The internal processes of an all-vanadium flow battery involve complex multi-physical field

coupling, encompassing the interplay of electrochemical reactions, thermal mass transport, and the transportation of fluids, electrons, ions, and heat across multiple physical domains. Therefore, studying the coupled transport characteristics of flow batteries is crucial for enhancing battery efficiency.

Although VFBs offer advantages in energy storage, they still have several limitations. Researchers, both domestically and internationally, have conducted extensive research in various areas, including material analysis, electrolyte transport processes, flow channels, and cell structural design. Numerical simulation methods [5] are widely utilized to analyze the electrochemical performance of all-vanadium flow batteries.

In terms of material analysis, graphite felt carbon [6], as the most commonly employed electrode material, has a well-established preparation and application system. However, its poor catalytic activity and low specific surface area [7] hinder further advancements in VFB technology. Enhancing the properties of graphite carbon felt material is a prominent research focus. Huang and Ma et al. developed a composite electrode material by modifying graphite felt [8], resulting in improved charging and discharging performance of the battery, achieving a current efficiency of over 96% at a current density of 300 A·m$^{-2}$. Kwon prepared a cost-effective mesoporous nitrogen-doped carbon structure using sodium citrate and urea precursors, which enhanced vanadium ion transfer and improved the performance of the redox reaction [9]. Additionally, significant attention has been given to the development and preparation of new diaphragms. Inadequate diaphragm materials may allow the vanadium ions to penetrate the membrane, leading to undesired side reactions. The challenge for researchers lies in providing reasonable proton conductivity while effectively suppressing vanadium ion crossover [10,11].

The losses caused by the electrolyte transport processes are also crucial, which is relative to the electrolyte flow, ion diffusion, variable operating conditions, and other aspects [12–14]. The electrolyte, as a crucial participant in the transport chain, serves as a site for storing and providing reactants. Its physical properties and transport process will definitely affect the performance of the VFB [15]. Firstly, the internal resistance caused by the mass transfer of the electrolytes will increase the ohmic overpotential. Thus, electrolytes with lower internal resistance and lower heat generation rates should be selected for higher system efficiency [16]. On the other hand, the flow rate of the electrolyte is a crucial factor to affect the overall performance of VFB. The power consumption of the circulating pump is positively related to the variation in flow rate, which affects the overall efficiency of VFB [17]. Ryan et al. conducted a study on the occurrence of side reactions when there was insufficient local flow supply. They proposed a novel method to regulate electrolyte flow utilizing a linear parameter adjustment framework [18]. The optimized flow rate can be obtained by maximizing the power output of VFB. Next, the electrolyte distribution accounts for another important factor for the VFB. The uniformity of the electrolyte is mainly affected by the geometric structure of the electrodes. What is more, the porosity of the electrode determines the transport performance of the ions in the electrolyte, which contributes to the losses of mass transfer and ohmic polarization [19–21]. At the macro level, the variation in electrolyte flow rate will cause a difference in local current, resulting in higher ohmic losses [22]. Ke X. et al. proposed an optimization principle of flow field design, which improved the flow rate [23]. Under high current density and power density, flow field design plays an important role in some large-scale energy storage stations [24]. Jienkulsawad et al. analyzed vanadium battery systems using different electrode and membrane materials [25]. Their aim was to investigate performance changes and capacity degradation caused by electrolyte imbalance. The results indicated that the use of higher-quality materials, increased reaction temperature, and higher vanadium ion concentration could mitigate the negative effects of electrolyte imbalance. Finally, for the operating conditions, changes in internal temperature also affect the transport characteristics of vanadium flow batteries, and this aspect cannot be ignored [26,27]. Praphulla R. and Ravendre G. et al. investigated the effect of low-temperature conditions on hydrodynamic parameters such as electrolyte viscosity and electrode permeability. The results showed

that as the operating temperature decreased, both viscosity and permeability increased, which led to additional power consumption. Therefore, achieving optimal performance at low temperatures requires a careful balance of parameters [28].

On the other hand, flow channel and cell structure design are also crucial factors influencing the performance of VFB [29]. Enhancing transport performance can be achieved by controlling electrolyte flow or optimizing the flow field structure, which can be comprehensively analyzed through simulation in conjunction with operating parameters and stack structure [30–32]. Huang et al. analyzed the flow field design process and flow optimization while also investigating the optimization methods and conducting a comprehensive comparison of cell performance with different flow field designs. The analysis revealed that a well-designed flow field could improve electrolyte distribution uniformity, enhance overall battery performance, and reduce usage costs [33]. Through the research of various scholars, it was found that compared to altering the flow, optimizing the flow field structure did not lead to a decrease in battery circulation system efficiency, making it a more desirable approach. Sun and Duan et al. conducted a numerical study on the impact of changes in electrode structure on all-vanadium flow battery performance [34]. The results indicated that both voltage efficiency and pressure drop increased with the introduction of various flow fields. Wang et al. compared the serpentine flow channel with the finger flow channel through simulation, proposing a reference strategy for flow field configuration optimization [35]. With the numerical method, Ehtesham Ali et al. conducted a study on a three-dimensional serpentine-type flow field with a different number of flow channels [36]. The results demonstrated that the pressure drop and pump power varied with the number of flow channels. Therefore, the flow channel structure was identified as one of the most important factors affecting battery performance. Existing studies have highlighted that the mass transfer process is crucial to the electrochemical reaction. Concentration polarization loss is a significant contributor to the efficiency loss in all-vanadium flow batteries [37]. Achieving a uniform distribution of reactants within the battery is essential [38] to reduce overall concentration polarization.

It can be seen that the cell structure was limited to rectangular, and little research is found in terms of novel cell structure. In this paper, a three-dimensional mathematical and physical model of the novel radial VFB cell was established based on our previous work [39]. We have studied the effect of the number of electrolyte inlets and the presence of electrolyte distribution passage on battery performance. The eight-inlet all-vanadium flow battery units have better performance, and the electrolyte distribution passage also significantly improves the battery performance. On this basis, the impact of cell thickness and operating parameters on the transport process was analyzed. The results will facilitate a broader application of the VFBs in new energy systems.

## 2. Model

### 2.1. Physical Model

For the novel circular VFB cell, as shown in Figure 1, the electrolyte enters from the outer side of the electrode and converges toward the center of the circle at the outlet. The sector-shape electrode is characterized by a gradual decrease in the flow area of the electrolyte from the inlet to the outlet. This feature impacts the distribution of activated ions, electrochemical reaction, and performance of the cell. In our previous work, the half-cell model of negative electrodes with different inlets was studied, and the model with eight inlets was recommended. Due to the geometric centrosymmetry of the circular VFB cell, the midline between the neighboring two inlets can be treated as a periodic interface. The simplified geometric model containing one inlet was established.

Figure 1 illustrates the circular electrode model with eight inlets. The detailed structural parameters of the battery unit are provided in Table 1. Thus, the recommended half-cell model of a negative electrode with eight inlets was taken as the original model in this paper.

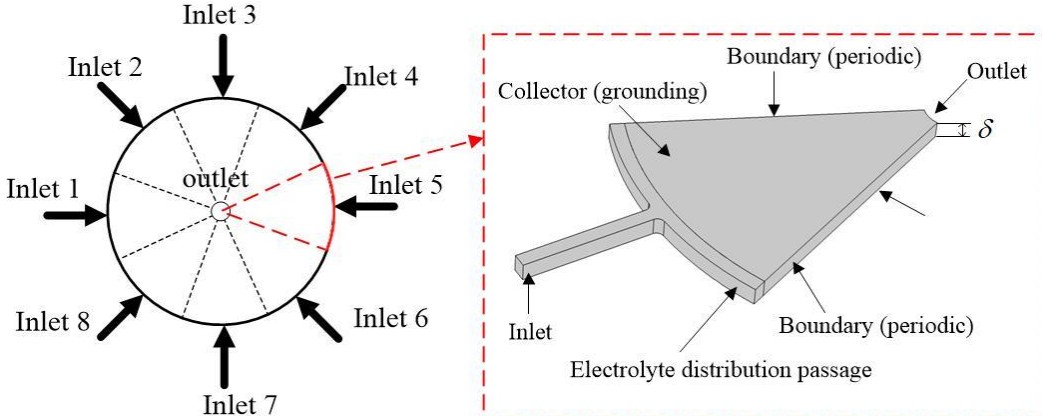

**Figure 1.** Schematic diagram of sector-shape VFB cell with 8 inlets.

**Table 1.** Cell structure parameters.

| Parameters | Symbols | Value | Unit |
|---|---|---|---|
| Electrode thickness | $\delta$ | 3.5 | mm |
| Inlet cross-sectional area | $S_{in}$ | 17.5 | mm$^2$ |
| Radial electrode inner radius | r | 10 | mm |
| Radial electrode outer radius | R | 112.5 | mm |

*2.2. Mathematical Model*

2.2.1. Assumptions

The electrolyte flow is assumed to be laminar and incompressible. The concentration of electrolyte ions is low, with water being the main component, so a dilute solution model is used. Furthermore, it is assumed that the physical properties of the electrolyte components are isotropic and homogeneous. The membrane and electrodes are also assumed to have isotropic and homogeneous physical properties. The penetration of vanadium ions within the membrane is not taken into account, and no side reactions are considered.

2.2.2. Electrochemical Reactions

The VFB cell comprises a proton exchange membrane, porous electrodes separated by the membrane, a current collection plate, the cathode and anode electrolyte tanks, and the circulation pipes. The electrolyte flows through the positive and negative porous electrodes and undergoes a redox electrochemical reaction to store or release electrical energy. The battery exclusively utilizes vanadium ions as the reactant, and the electrolyte is composed of a sulfuric acid solution containing vanadium ions of different valence. The $V^{4+}/V^{5+}$ and $V^{2+}/V^{3+}$ redox pairs are stored in the positive and negative electrolyte tanks, respectively [40]. The reactions that take place during the charging and discharging process are as follows:

Positive:

$$VO^{2+} + H_2O - e^- \rightleftharpoons VO_2^+ + 2H^+ \tag{1}$$

Negative:

$$V^{3+} + e^- \rightleftharpoons V^{2+} \tag{2}$$

2.2.3. Governing Equations

The mass conservation equations for the various substances during an electrochemical reaction are described as follows:

$$\vec{v}\nabla c_i - D_i^{eff}\nabla^2 c_i = S_i \tag{3}$$

where $v$ is the flow rate; $c_i$ represents the concentration of substance I; $S_i$ represents the source term of substance i, as defined in Table 2, and $D_i^{eff}$ is the effective diffusion coefficient, obtained from the diffusion coefficient of the substance via the following Bruggemann correction:

$$D_i^{eff} = \varepsilon^{\frac{3}{2}} D_i \tag{4}$$

where $\varepsilon$ is the porosity of the electrode, and from the law of conservation of charge, it follows that

$$\vec{i}_s = \vec{i}_e = S_\phi \tag{5}$$

where $S_\phi$ denotes the source term for the conservation of charge, defined in Table 2, and $i_e$ and $i_s$ denote, respectively, the current densities of ions and electrons obtained by Ohm's law:

$$\vec{i}_s = -\sigma_s^{eff} \nabla \phi_s \tag{6}$$

$$\vec{i}_e = -\kappa_1 \nabla \phi_1 \tag{7}$$

where $\phi_s$ and $\phi_1$ denote the electron and ion potentials, respectively, and $\sigma_s^{eff}$ is the effective electronic conductivity of the porous electrode, which is related to the properties of the material and is obtained via the following Bruggemann correction:

$$\sigma_s^{eff} = (1 - \varepsilon)^{\frac{3}{2}} \sigma^s \tag{8}$$

where $\kappa_1$ is the ionic conductivity and is obtained by the following equation:

$$\kappa_1 = 35.716 + 7.699 \times SOC \tag{9}$$

where $SOC$ is the state of charge.

**Table 2.** Source items.

| Source | | Negative | |
|---|---|---|---|
| $S_i$ | $V^{2+}$ | a*j/F | |
| | $V^{3+}$ | −a*j/F | |
| $S_\phi$ | $\phi_s$ | a*j | |
| | $\phi_1$ | −a*j | |

The Butler–Volmer law is sufficiently accurate in describing reversible redox reactions on porous electrode surfaces so that the transfer current density can be expressed as follows:

$$j = i_c^0 \left[ \frac{c_{V^{3+}}^S}{c_{V^{3+}}} exp\left( -\frac{\alpha_c F \eta_2}{RT} \right) - \frac{c_{V^{2+}}^S}{c_{V^{2+}}} exp\left( \frac{\alpha_a F \eta_2}{RT} \right) \right] \tag{10}$$

where $\alpha_a$ and $\alpha_c$ are the anodic and cathodic transfer coefficients, respectively; $ci^S$ is the concentration of the substance at the solid–liquid interface; $i_c^0$ is the exchange current density, and $k_c$ is the standard rate number of the following reaction:

$$i_c^0 = Fk_c \left( c_{V^{2+}} \right)^{\alpha_c} \left( c_{V^{3+}} \right)^{\alpha_a} \tag{11}$$

The positive and negative electrode overpotentials are expressed as follows:

$$\eta_1 = \phi_s - \phi_1 - E_1 \tag{12}$$

$$\eta_2 = \phi_s - \phi_1 - E_2 \tag{13}$$

where $E_1$ and $E_2$ are the equilibrium potentials for the positive and negative redox reactions, respectively, expressed by the following Nernst equation:

$$E_1 = E_1^0 + \frac{RT}{F}\ln\left(\frac{c_{V^{5+}} \times (c_{H^+})^2}{c_{V^{4+}}}\right) \tag{14}$$

$$E_2 = E_2^0 + \frac{RT}{F}\ln\left(\frac{c_{V^{3+}}}{c_{V^{2+}}}\right) \tag{15}$$

where $E_1{}^0$ and $E_2{}^0$ are the positive and negative standard potentials, respectively; $F$ is the Faraday constant; $R$ is the ideal gas constant, and the cell voltage is calculated as follows:

$$E_{\text{cell}} = E_1 - E_2 - \eta_1 - \eta_2 - IAR_{\text{cell}} \tag{16}$$

where $I$ is the current density; $A$ is the surface area of the electrode, and $R_{\text{cell}}$ is the resistance.

For the half-cell model, the electrode potential is calculated as follows:

$$E_{\text{ele}} = \phi_1 - \phi_s - IAR_{\text{neg}} \tag{17}$$

where $R_{\text{neg}}$ means the negative electrode resistance.

The stationary incompressible flow in the inlet and outlet pipes is usually represented by the following Navier–Stokes equation:

$$\rho\left(\vec{v}\cdot\nabla\right)\vec{v} = -p + \mu\left[\nabla\vec{v} + \left(\nabla\vec{v}\right)^T\right] \tag{18}$$

$$\nabla\cdot\vec{v} = 0 \tag{19}$$

The velocity $v$ in porous media is given by Darcy's law and the following Coetzee–Kalman equation:

$$\vec{v} = \frac{d_f^2}{\kappa\mu}\cdot\frac{\varepsilon^3}{(1-\varepsilon)^2}\nabla^2 p = 0 \tag{20}$$

where $d_f$ is the mean fiber diameter; $p$ is the liquid pressure; $\mu$ is the dynamic viscosity of the liquid, and $\kappa$ is the Kozeny–Carman constant, and the following pressure relationship can be obtained based on the assumption of the approximate dilute solutions and incompressibility:

$$-\frac{d_f^2}{\kappa\mu}\cdot\frac{\varepsilon^3}{(1-\varepsilon)^2}\nabla^2 p = 0 \tag{21}$$

The power loss $P_{\text{loss}}$ of the cell is defined as follows, incorporating each concentration polarization within the porous electrode, the ohmic losses, and pump power losses; $\Delta u_{\text{men}}$ is the ion exchange membrane pressure drop; $L_{\text{men}}$ is the ion exchange membrane thickness; $Q^0$ is the imported electrolyte flow; $p$ is the pressure drop, and $\varphi_P$ is the pump efficiency.

$$P_{\text{loss}} = IA(\eta_1 + \eta_2) + (\Delta u_{\text{men}})^2\cdot\frac{A\sigma_{\text{men}}}{L_{\text{men}}} + Q^0\cdot\frac{\Delta p}{\varphi_P} \tag{22}$$

The total battery power $P_{\text{total}}$ is defined as follows, where $P_{\text{net}}$ is the actual net power output from the battery:

$$P_{\text{total}} = P_{\text{net}} + P_{\text{loss}} \tag{23}$$

The energy efficiency $\psi_{\text{power}}$ is defined as follows:

$$\psi_{\text{power}} = \frac{P_{\text{net}}}{P_{\text{total}}} \tag{24}$$

### 2.2.4. Boundary Conditions

This analysis primarily focuses on the negative half-cell, and the simulation is conducted under multi-physics field conditions. The boundary condition of grounding is applied to the collector plate boundary:

$$\phi_s = 0 \tag{25}$$

$$-\kappa_1 \nabla \phi_1 \cdot \vec{n} = I \tag{26}$$

where Equation (26) is the electrolyte current density flux condition setting for the membrane boundary, chosen as the outward unit normal vector, and $I$ is the applied current density.

Determine the inlet vanadium ion concentration at the negative side of the battery based on *SOC* and initial vanadium ion concentration:

$$c_{V^{2+}}^{in} = c^0 \cdot SOC \tag{27}$$

$$c_{V^{3+}}^{in} = c^0 \cdot (1 - SOC) \tag{28}$$

According to the setting of the total electrolyte flow at the electrolyte inlet, the velocity values are defined according to the different working conditions. The electrolyte outlet is set as the pressure outlet boundary. The radial boundary at both sides of each sector electrode unit is set as the period boundary conditions. All other boundaries are defined as flux-free and no-slip boundary conditions.

### 2.2.5. Battery Performance Parameters

The modeling is performed in COMSOL, and the solution is combined with the finite volume element method. Combinations of different interfaces are used to solve convective diffusion equations, differential equations, etc. The operating parameters of the battery are shown in Table 3, and the performance design parameters of the battery are shown in Table 4.

**Table 3.** Operating parameters.

| Parameters | Symbols | Value | Unit |
|:---:|:---:|:---:|:---:|
| Temperature | $T$ | 298 | K |
| Electrolyte flow | $Q$ | 663.6 | ml/min |
| Outlet pressure | $P_{out}$ | 0 | Pa |
| Electrode current density | $I$ | 1600 | A/m$^2$ |
| State of Charge | *SOC* | 80 | % |

### 2.3. Grid Independence Verification

Meshing plays a crucial role in the simulation process. In this study, COMSOL is employed for meshing the cell model, and a free quadrilateral mesh is selected for refinement to ensure the accuracy of the model. The aim is to achieve a result that closely approximates the real solution by choosing an appropriate computational accuracy. The grid independence of the model is verified using the eight-inlet model with distribution pipes. The change in the average concentration of trivalent vanadium ions within the cell is compared for different grid sizes, as shown in Table 5. It is observed that as the number of grids increases from 61, 893 to 86, 532, the change in the average concentration of $V^{3+}$ is lower than 1%. Therefore, a grid number of 61,893 is chosen as it provides sufficient accuracy for the analysis.

**Table 4.** Battery performance design parameters.

| Parameters | Symbols | Value | Unit | Source |
|---|---|---|---|---|
| Porosity | $\varepsilon_p$ | 0.929 | - | Literature [41] |
| Specific surface area | $a$ | $1.62 \times 10^4$ | $m^{-1}$ | Literature [41] |
| Carbon fiber diameter | $d_p$ | $1.76 \times 10^{-5}$ | m | Literature [41] |
| Electrical conductivity | $\sigma_s$ | 1000 | S/m | Literature [42] |
| Kornitze–Kalman coefficient | $k_{CK}$ | 4.28 | - | Literature [42] |
| Viscosity | $M$ | $4.928 \times 10^{-3}$ | Pa·s | Literature [42] |
| Initial proton concentration at the negative electrode | $C_{H\_0\_neg}$ | 4500 | $mol/m^3$ | Literature [42] |
| Cathodic transfer coefficient | $\alpha_c$ | 0.5 | - | Literature [42] |
| Anode transfer coefficient | $\alpha_a$ | 0.5 | - | Literature [42] |
| Diffusion coefficient of $V^{2+}$ | $D_{V2}$ | $2.4 \times 10^{-10}$ | $m^2/s$ | Literature [43] |
| Diffusion coefficient of $V^{3+}$ | $D_{V3}$ | $2.4 \times 10^{-10}$ | $m^2/s$ | Literature [43] |
| Initial concentration of water | $C_{H2O}$ | 46,500 | $mol/m^3$ | Literature [44] |
| Negative standard reaction rate constant | $k_{neg}$ | $1.7 \times 10^{-7}$ | $m^2/s$ | Literature [45] |
| Standard equilibrium potential | $E_{eq}$ | $-0.255$ | V | Literature [46] |

**Table 5.** Grid independence verification.

| Number of Grid Cells | 42,556 | 61,893 | 86,532 |
|---|---|---|---|
| Average ion concentration of $V^{3+}$ ($mol/m^3$) | 259.6 | 263.2 | 265.8 |

*2.4. Model Validation*

In reference [47], a numerical model was developed to investigate the sector cell structure with the geometric configuration depicted in Figure 2a. The velocity field distribution inside the cell was determined and presented in Figure 2c. By utilizing the same geometric and cell performance parameters as the reference study, along with the mathematical model and simulation approach employed in this paper, the velocity distribution of the electrolyte was obtained and found to be consistent with the reference study. The electrode thickness, $\delta$, was known to be 3.5 mm. The velocity distribution of the electrolyte in the plane of y = $\delta$/2 = 1.75 mm was calculated and plotted, as shown in Figure 2b. It can be observed that the distribution of the electrolyte flow rate was largely in agreement with the reference study. Additionally, the average flow rate of the electrolyte within the model was calculated to be 0.0171 m/s, which falls within an acceptable range when compared to the value of 0.0185 m/s reported in the literature. Consequently, the accuracy of the numerical simulation model in this paper is indirectly validated.

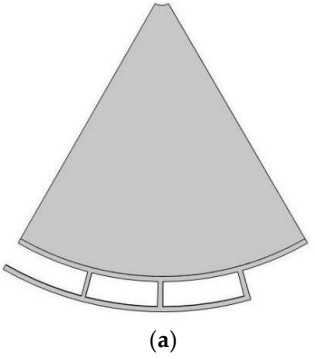

(a)

**Figure 2.** *Cont.*

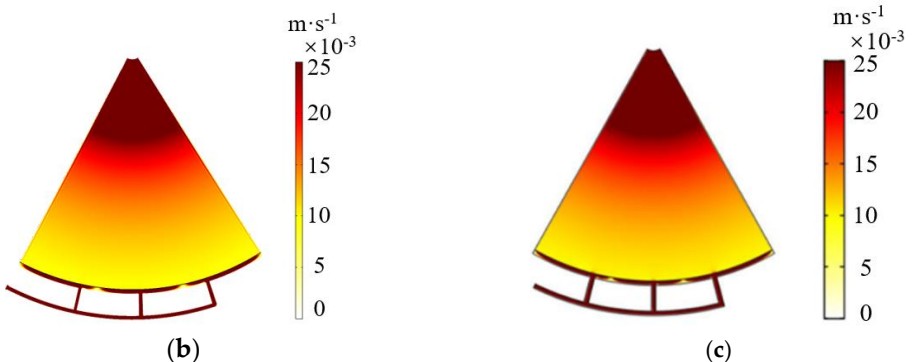

**(b)**                                                     **(c)**

**Figure 2.** Comparison of electrolyte speed distribution. (**a**) Sector cell structure; (**b**) Velocity distribution in this paper; (**c**) Velocity distribution in literature [47].

## 3. Results and Discussions

### 3.1. Electrode Thickness Optimization Analysis

The thickness $\delta$ of the negative electrode has a significant impact on the electrochemical performance of the VFB. The eight-inlet model with distribution passage, recommended in the literature [30], has been verified and adopted as the basis for all the research in this paper. Electrode models with thicknesses ranging from 1 mm to 9 mm with an interval of 1 mm were established. For various electrode thicknesses, the design operating parameters, such as temperature, electrolyte concentration, and electrolyte flow, were kept constant. For brevity, the visualized results for different electrode thicknesses of 1 mm, 3 mm, 5 mm, and 7 mm are presented and analyzed.

#### 3.1.1. Variations

The state of charge (SOC) represents the overall charge level of the cell and is often associated with the initial electrolyte concentration. In this simulation, the SOC value was set at 0.8. For different electrode thicknesses, Figures 3 and 4 illustrate the electrolyte pressure and $V^{3+}$ concentration distributions in the middle section of the negative electrode. It can be observed that the distributions of the electrolyte pressure and $V^{3+}$ concentration are homogeneous along the circumferential direction due to the effect of electrolyte distributing passage, while the pressure decreases and the concentration increases along the sector-shape direction. Numerically, pressure drop varies in the order of $6.4 \times 10^4$ Pa, $2.3 \times 10^4$ Pa, $1.4 \times 10^4$ Pa and $0.9 \times 10^4$ Pa. The $V^{3+}$ concentration is in the order of 330.8 mol/m$^3$, 319.97 mol/m$^3$, 314.9 mol/m$^3$, and 314.3 mol/m$^3$. Overall, the average electrolyte pressure decreases as the thickness increases. This is because, as the thickness increases, the cross-sectional area for electrolyte flow also increases, resulting in reduced flow resistance. The average $V^{3+}$ concentration follows a general downward trend. This trend is due to the fact that as the thickness increases, the battery can provide more reaction sites for electrochemical reactions, which accelerates the consumption of $V^{3+}$. However, the total incoming flow of electrolyte is kept constant, so the electrolyte does not fully fill the electrode in a timely manner, which may result in an undesirable distribution for some thicker configurations.

#### 3.1.2. Performance Parameters

The overpotential, which can be affected by factors such as active polarization loss and mass transfer loss, was analyzed as an important performance parameter. The variation in the absolute value of the overpotential was calculated and presented in Figure 5. It can be observed that the absolute value of the overpotential decreases by 0.76 V. Therefore, thicker electrodes correspond to lower overpotential, which indicates a better battery performance. However, when considering the electrode potential, the results are completely opposite. Figure 6 illustrates the change in electrode potential with electrode thickness ranging from 1 mm to 9 mm. The electrode with a thickness of 1 mm exhibits the highest potential of

2.992 V, which is 2.56 V higher than the potential of 0.432 V at 9 mm. The potential shows a decreasing trend, which is detrimental to battery performance. This is because an increase in electrode thickness leads to an increase in internal resistance, resulting in more ohmic losses. Additionally, the increase in electrode thickness also leads to an increase in the cross-sectional area of the electrolyte flow. With a constant incoming flow of electrolytes, the flow rate becomes slower, affecting the electrode potential. From the perspective of electrode potential, a smaller electrode thickness value indicates better battery performance.

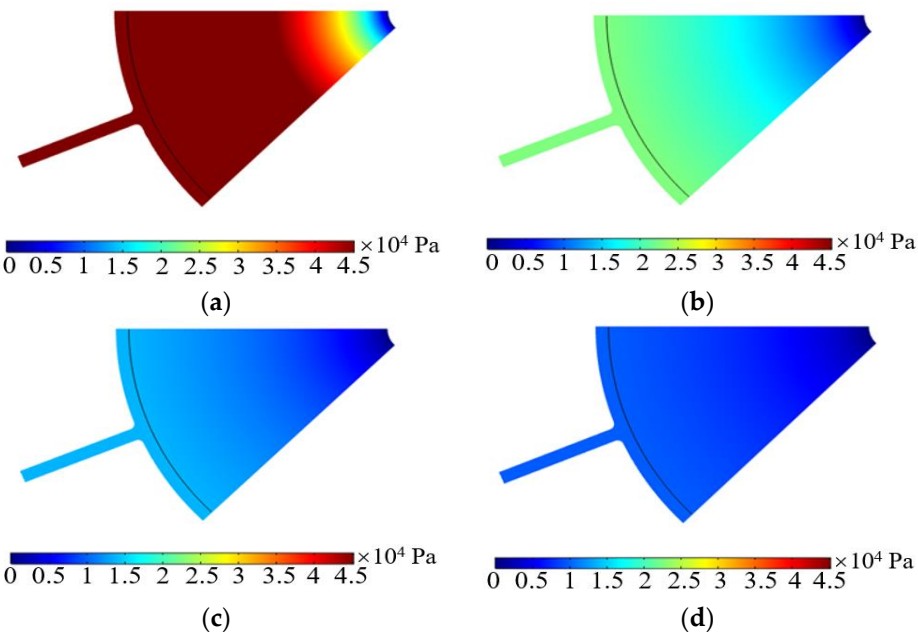

**Figure 3.** Pressure distribution at different electrode thicknesses. (**a**) The thickness of 1 mm. (**b**) The thickness of 3 mm. (**c**) The thickness of 5 mm. (**d**) The thickness of 7 mm.

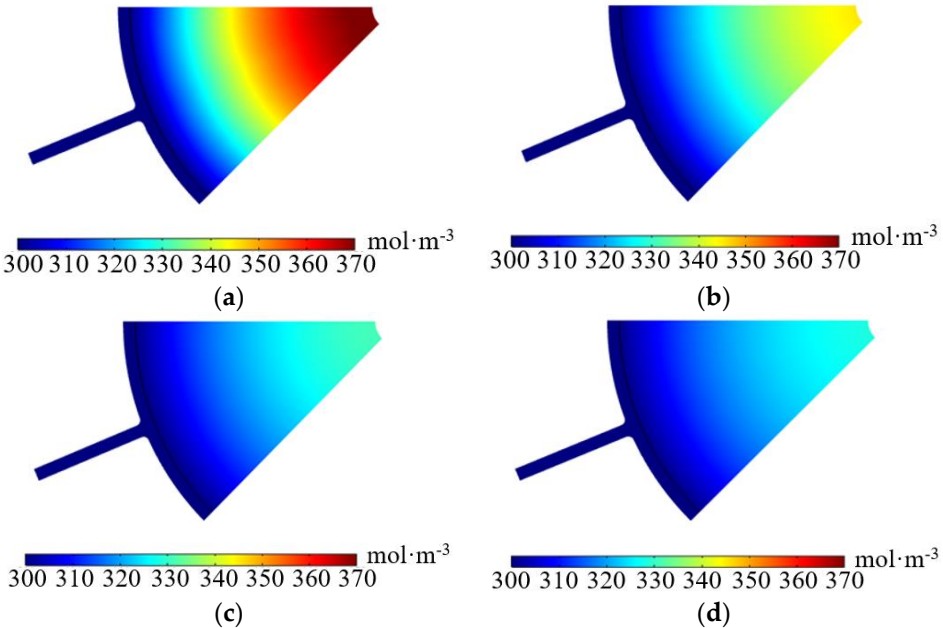

**Figure 4.** Concentration of $V^{3+}$ distribution at different electrode thicknesses. (**a**) The thickness of 1 mm. (**b**) The thickness of 3 mm. (**c**) The thickness of 5 mm. (**d**) The thickness of 7mm.

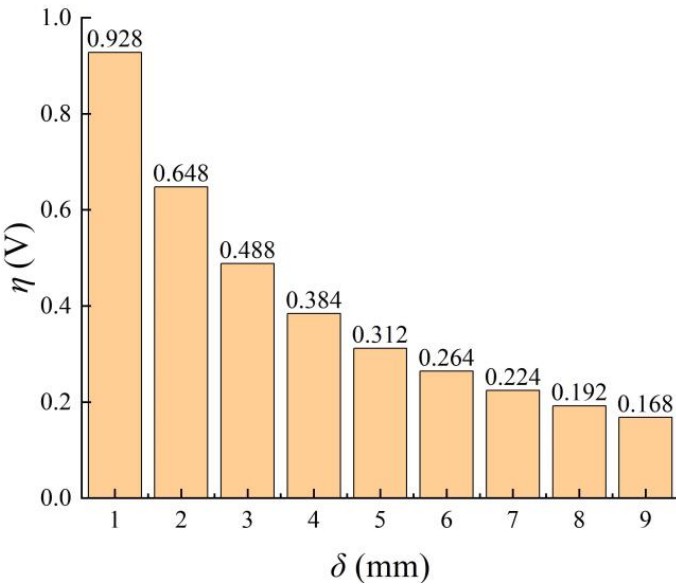

**Figure 5.** Variation in the absolute value of the overpotential at different electrode thicknesses.

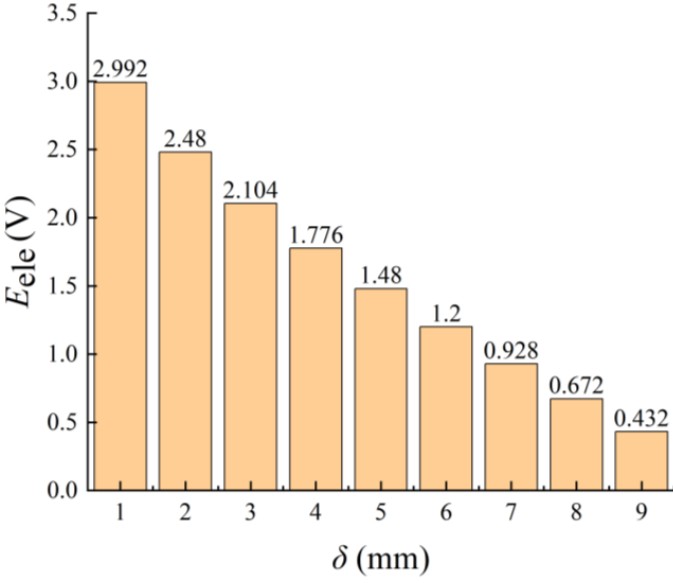

**Figure 6.** Variation in potential at different electrode thicknesses.

In general, from the perspective of energy conservation, energy efficiency is adopted to evaluate the overall performance of the battery cell with different electrode thicknesses. The variation in energy efficiency is influenced by the actual output power and the power losses. Based on the previous analysis, it is observed that as the electrode thickness increases, the electrode potential decreases, resulting in a decrease in net power. However, the overpotential also decreases, leading to a decrease in power losses. Thus, the biggest energy efficiency will occur at a specific electrode thickness. As a result, the energy efficiency is calculated for different electrode thicknesses, as shown in Figure 7. It can be observed that as the electrode thickness increases, the energy efficiency initially increases and then decreases. The maximum energy efficiency achieved is 0.8161, corresponding to an optimal electrode thickness of 5 mm. Therefore, the optimal electrode thickness of 5 mm is promoted for the operating characteristics analysis.

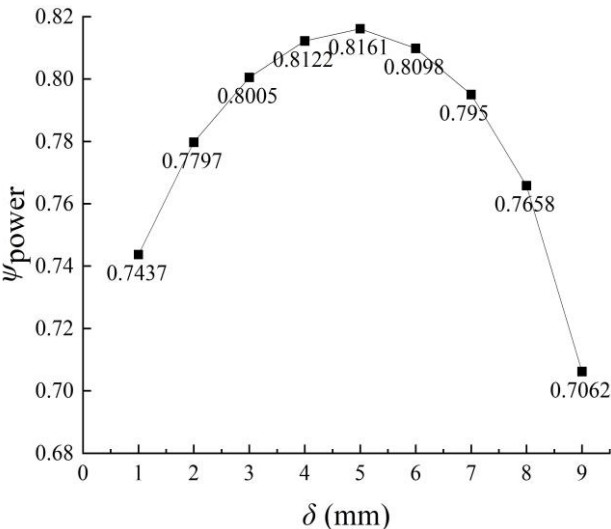

**Figure 7.** Variation in energy efficiency at different electrode thicknesses.

*3.2. Operating Characteristics Analysis*

The electrolyte serves as the conductive material in the VFB cell and plays a crucial role in energy storage and conversion. The ions in the electrolyte act as the reactants for electrochemical reactions occurring on the electrode surfaces. Therefore, the properties of the electrolyte are the most important factors affecting battery performance. The electrode current density is another important parameter that affects battery performance. It impacts the charging and discharging cycles, self-discharge phenomenon, and so on. Based on the previously promoted half-cell model with a thickness of 5 mm, simulations are conducted under different operating conditions, including the electrolyte flow, electrolyte concentration, and current density. The results of the electrode overpotential, concentration of the $V^{3+}$, electrode potential, and energy efficiency are presented and analyzed.

3.2.1. Imported Electrolyte Flow Rate

The influence of the operating parameter called imported electrolyte flow rate on the electrochemical characteristics of the VFB cell is studied in this part of this paper. The imported electrolyte flow rates considered are 464 mL/min, 564 mL/min, 664 mL/min, 764 mL/min, and 864 mL/min. Other operating parameters are kept consistent. At first, the contours of the electrolyte pressure and $V^{3+}$ concentration are presented and analyzed, and then, the performance parameters, including the electrode overpotential and electrode potential, are calculated and analyzed. Finally, the overall performance evaluating parameter called energy efficiency is obtained.

The distribution of electrolyte pressure under different imported electrolyte flow rates is shown in Figure 8. It can be observed that the pressure exhibits a uniform distribution along the circumferential direction. The pressure drop in electrolyte flowing through the electrode increases in the following order: $1.3 \times 10^4$ Pa, $1.6 \times 10^4$ Pa, $1.9 \times 10^4$ Pa, and $2.2 \times 10^4$ Pa, showing a gradual increase. This trend indicates that a greater pressure drop results in higher pump power consumption at a higher electrolyte flow rate.

From Figure 9, it is evident that the values of the average concentration of $V^{3+}$ vary in the order of 328.29 mol/m$^3$, 337.96 mol/m$^3$, 344.67 mol/m$^3$, and 349.61 mol/m$^3$, increasing steadily with the imported electrolyte flow rate increasing from 464 mL/min to 764 mL/min. This is because a higher imported electrolyte flow rate ensures an ample supply of reactants within the electrode, thereby reducing the impact of concentration polarization on battery performance. Although the concentration values continue to increase steadily, the rate of increase diminishes. Therefore, the effect of changes in electrolyte flow rate on $V^{3+}$ concentration is limited. The reason for this limitation is that while the electrolyte flow continues to increase, the volume of the electrode remains constant. As a result, the effective

reaction area provided by the porous electrode reaches its limit. Some of the electrolyte is expelled from the electrode without undergoing a reaction. Thus, it can be observed that when the imported electrolyte flow reaches a certain value, its impact on ion concentration gradually levels off.

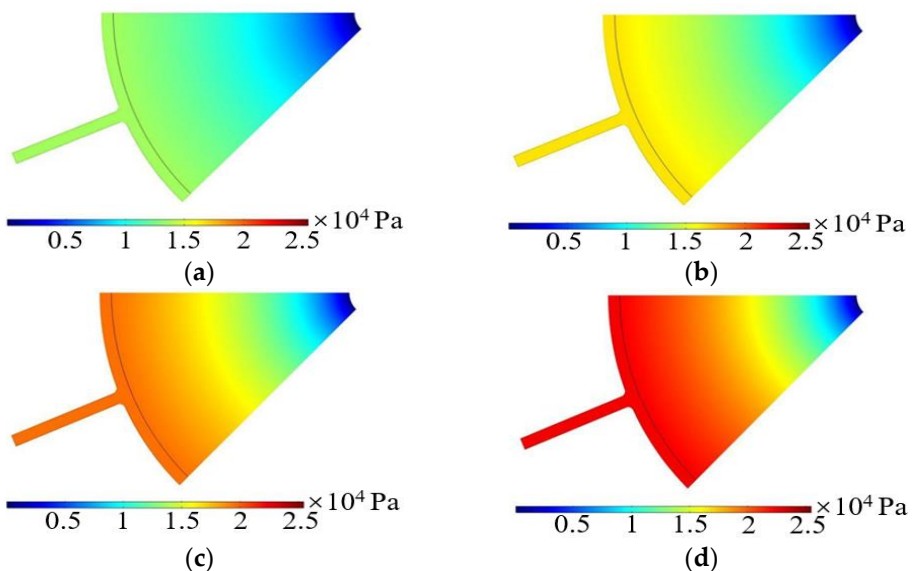

**Figure 8.** Pressure distribution at different electrolyte flow rates. (**a**) 464 mL/min. (**b**) 564 mL/min. (**c**) 664 mL/min. (**d**) 764 mL/min.

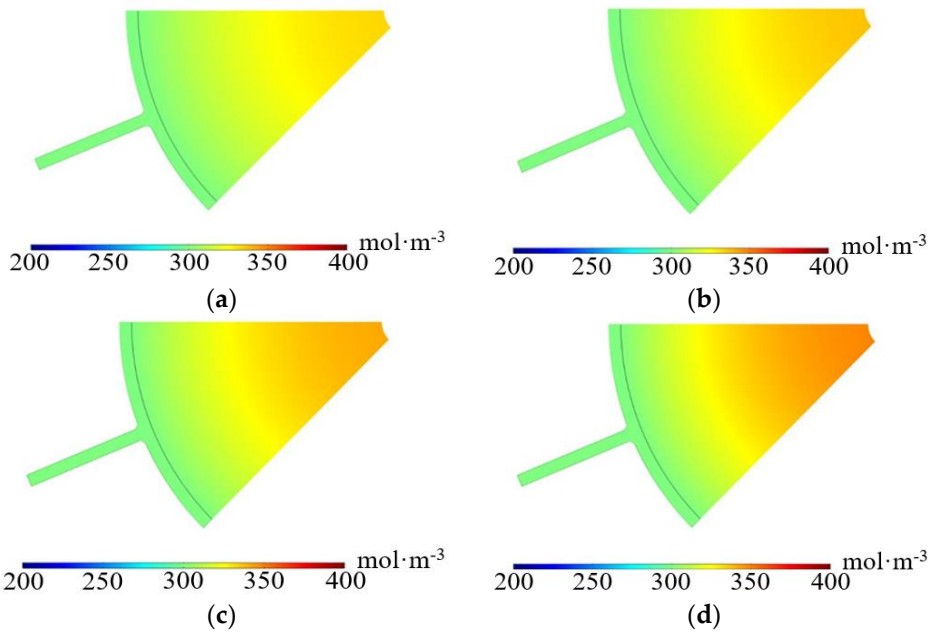

**Figure 9.** Concentration distribution of $V^{3+}$ at different electrolyte flow rates. (**a**) 464 mL/min. (**b**) 564 mL/min. (**c**) 664 mL/min. (**d**) 764 mL/min.

Figure 10 displays the variation in the absolute value of overpotential with changes in imported electrolyte flow. It can be observed that the absolute value of overpotential decreases from 0.437 V to 0.424 V, indicating a decreasing trend. Figure 11 illustrates the potential variation from 464 mL/min to 864 mL/min for the total imported electrolyte flow. It shows a steady increase in potential. From these results, it can be inferred that higher imported electrolyte flow leads to better battery performance. This is because the mass transfer within the electrode is enhanced, thereby reducing the impact of concentration polarization. However, both the increase in potential and the decrease in overpotential slow

down. This is attributed to the fact that the effective reaction area provided by the porous electrode reaches its limit. As a result, further improvements in performance are limited.

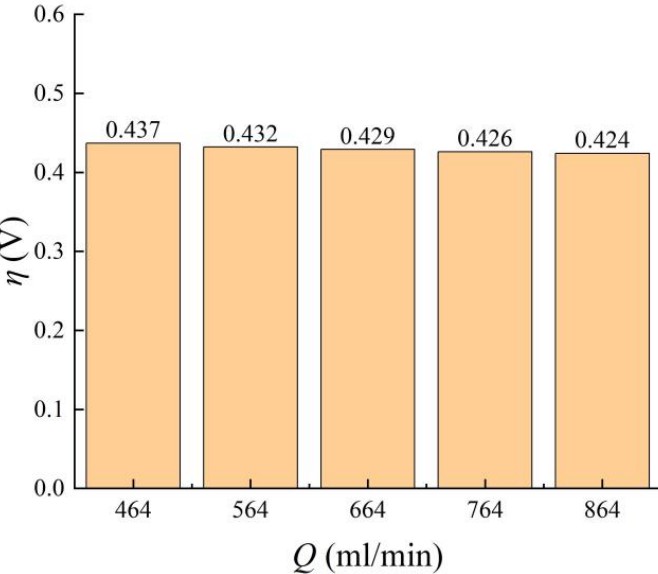

**Figure 10.** Variation in the absolute value of overpotential with imported electrolyte flow.

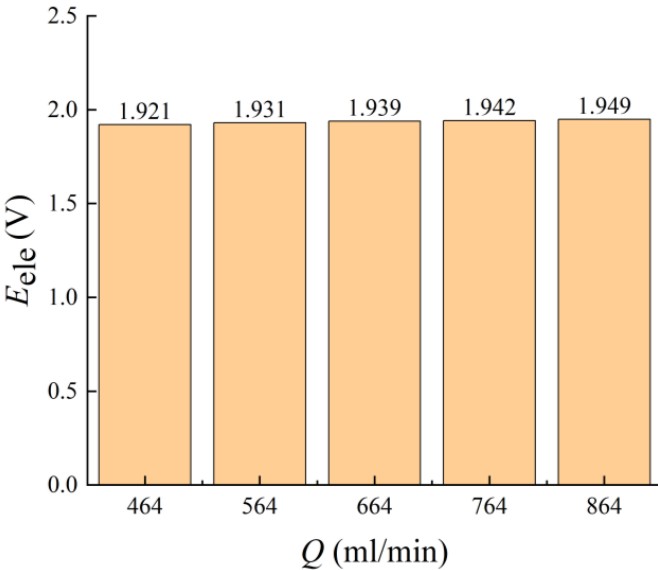

**Figure 11.** Variation in potential with imported electrolyte flow.

Figure 12 presents the variation in energy efficiency with changes in imported electrolyte flow. As the imported electrolyte flow increases, the volume of the electrode remains limited. Consequently, the effective reaction area provided by the porous electrode reaches its limit, resulting in some electrolytes being expelled from the electrode without undergoing a reaction. This leads to losses in the system. Furthermore, the increase in imported electrolyte flow requires more pump power, resulting in additional pump power losses. Consequently, the battery power loss increases with the increase in electrolyte flow. The lowest power loss in the battery is observed at 3.457 W for an electrolyte flow of 464 mL/min. Although the net power increases with the increase in potential, it does not compensate for the effect of power loss. As a result, the energy efficiency decreases with the increase in imported electrolyte flow.

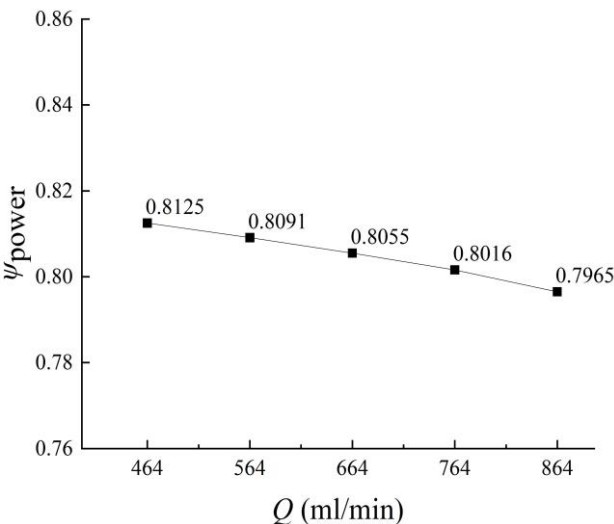

**Figure 12.** Variation in energy efficiency with imported electrolyte flow.

### 3.2.2. Imported Electrolyte Concentration

The calculations were initially performed for different electrolyte concentration operating conditions, with a fixed SOC value of 0.8. The initial vanadium concentrations of 1100 mol/m$^3$, 1300 mol/m$^3$, 1500 mol/m$^3$, 1700 mol/m$^3$, and 1900 mol/m$^3$ were considered, while keeping other parameters consistent. The results are analyzed as follows.

Figures 13 and 14 illustrate the pressure distribution and V$^{3+}$ concentration distribution at different imported electrolyte concentrations. It can be observed that the pressure distribution does not exhibit significant changes with varying initial electrolyte concentrations. However, the average V$^{3+}$ concentration values vary in the order of 257.1 mol/m$^3$, 296.3 mol/m$^3$, 336.8 mol/m$^3$, and 377.2 mol/m$^3$. The results demonstrate a steady upward trend. This is because a higher concentration of electrolyte provides sufficient reactants and ensures strong material transport within the electrolyte. Additionally, it improves the uniformity of concentration distribution. Therefore, from the perspective of material transport and concentration distribution, increasing the initial electrolyte concentration is beneficial for enhancing electrode performance.

The absolute values of overpotential at different imported electrolyte concentrations were calculated and analyzed, as shown in Figure 15. As the initial concentration increased, the absolute value of overpotential decreased from 0.529 V to 0.364 V. The results demonstrate a gradual downward trend, with the absolute value of overpotential at the optimum concentration being approximately 31.2% lower than that at the lowest concentration. This reduction in overpotential is primarily attributed to the decrease in mass transfer losses. The higher electrolyte concentration ensures stronger material transport within the electrode, leading to an increased uniformity of concentration distribution and a subsequent decrease in overpotential. Figure 16 illustrates the changes in electrode potential. It can be observed that the electrode at 1900 mol/m$^3$ exhibits the highest potential of 2.027 V, which is 0.152 V higher than the potential of 1.875 V at 1100 mol/m$^3$. Therefore, based on the above results, it can be concluded that a higher electrolyte concentration yields better cell performance under the same operating parameters and cell configuration.

The energy efficiency of the VFB cell under different imported electrolyte concentrations is calculated and presented in Figure 17. It can be observed that the energy efficiency increases with an increase in the concentration of the imported electrolyte. This is attributed to the more uniform concentration and distribution of reactants, leading to more efficient reactions and reduced polarization losses. Additionally, the pump power loss remains relatively constant as the electrolyte flow rate remains constant, and the pressure drop does not significantly change. Moreover, the potential increases, resulting in an increase in

net power. Therefore, the combined effect of these factors leads to an overall increase in energy efficiency.

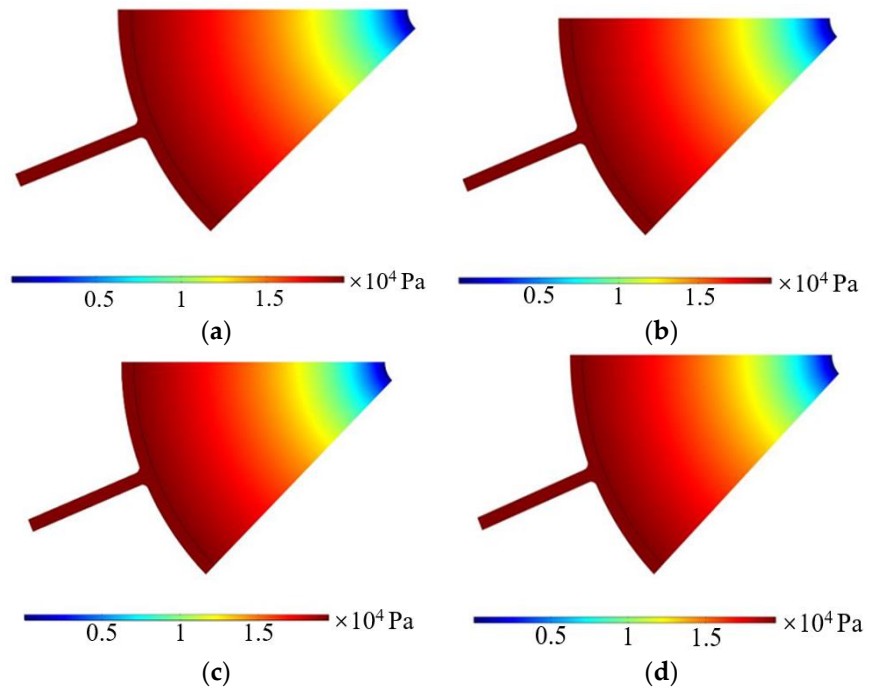

**Figure 13.** Pressure distribution at different electrolyte concentrations. (**a**) Concentration of 1100 mol/m$^3$. (**b**) Concentration of 1300 mol/m$^3$. (**c**) Concentration of 1500 mol/m$^3$. (**d**) Concentration of 1700 mol/m$^3$.

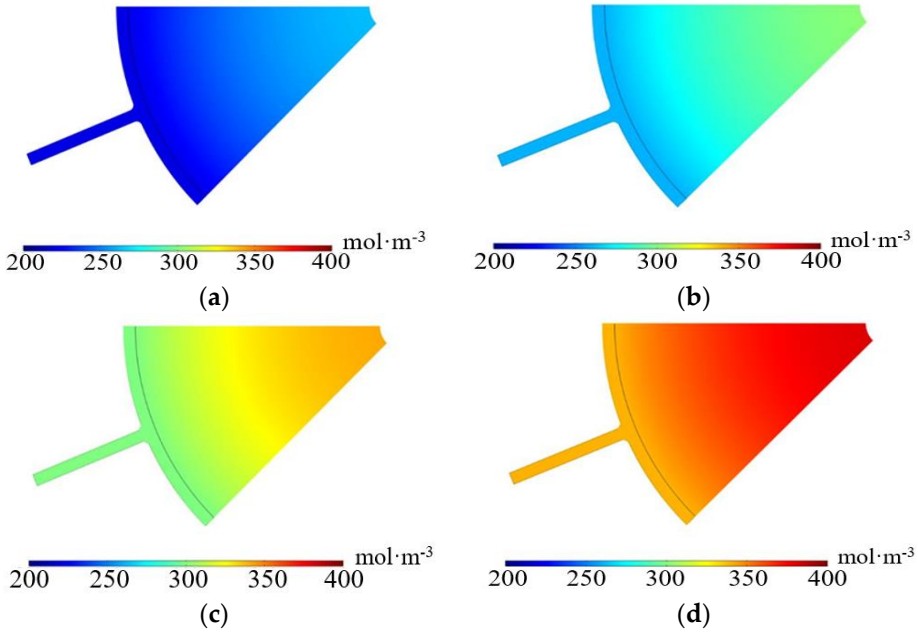

**Figure 14.** Concentration distribution of V$^{3+}$ at different electrolyte concentrations. (**a**) Concentration of 1100 mol/m$^3$. (**b**) Concentration of 1300 mol/m$^3$. (**c**) Concentration of 1500 mol/m$^3$. (**d**) Concentration of 1700 mol/m$^3$.

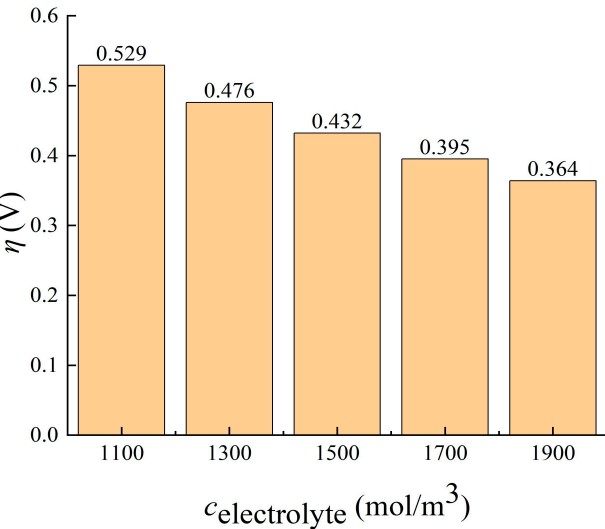

**Figure 15.** Variation in absolute value of electrode overpotential with electrolyte concentration.

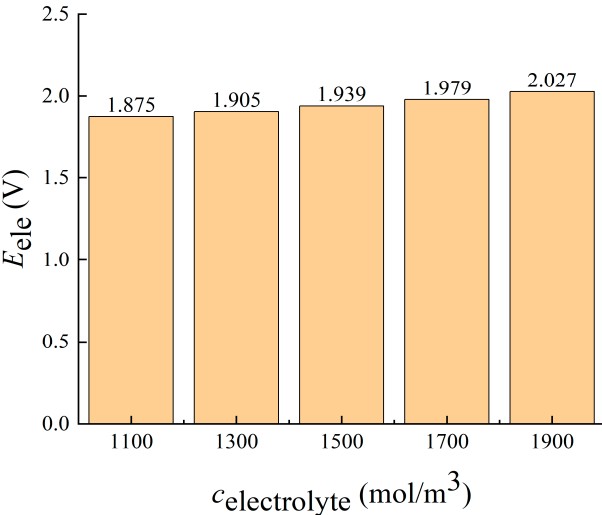

**Figure 16.** Variation in potential with electrolyte concentration.

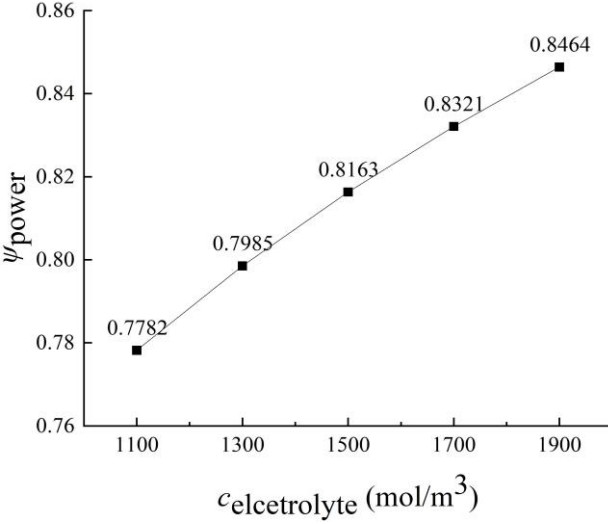

**Figure 17.** Variation in energy efficiency with electrolyte concentration.

### 3.2.3. Electrode Current Density

This analysis is now conducted for different electrode current densities, with a controlled temperature of 25 °C and an initial electrolyte concentration of 1500 mol/m$^3$. The electrode current densities considered are 1200 A/m$^2$, 1400 A/m$^2$, 1600 A/m$^2$, 1800 A/m$^2$, and 2000 A/m$^2$, while keeping other parameters consistent. The results are analyzed as follows.

Figures 18 and 19 illustrate the pressure and trivalent vanadium ion concentration distributions, respectively. It can be observed that the pressure distribution does not exhibit significant variations with changes in current density. The concentration distribution appears to be relatively uniform. The values of average V$^{3+}$ vary in the order of 327.09 mol/m$^3$, 331.98 mol/m$^3$, 336.99 mol/m$^3$, and 342.11 mol/m$^3$, which shows an increase at different electrode current densities. The results demonstrate an increase in V$^{3+}$ concentration with increasing the electrode current density. It can be attributed to the fact that higher current densities reduce the charging and discharging time of the cell, and the conversion of chemical energy into electrical energy is accelerated. As a result, the concentration of trivalent vanadium in the negative electrolyte increases during discharge.

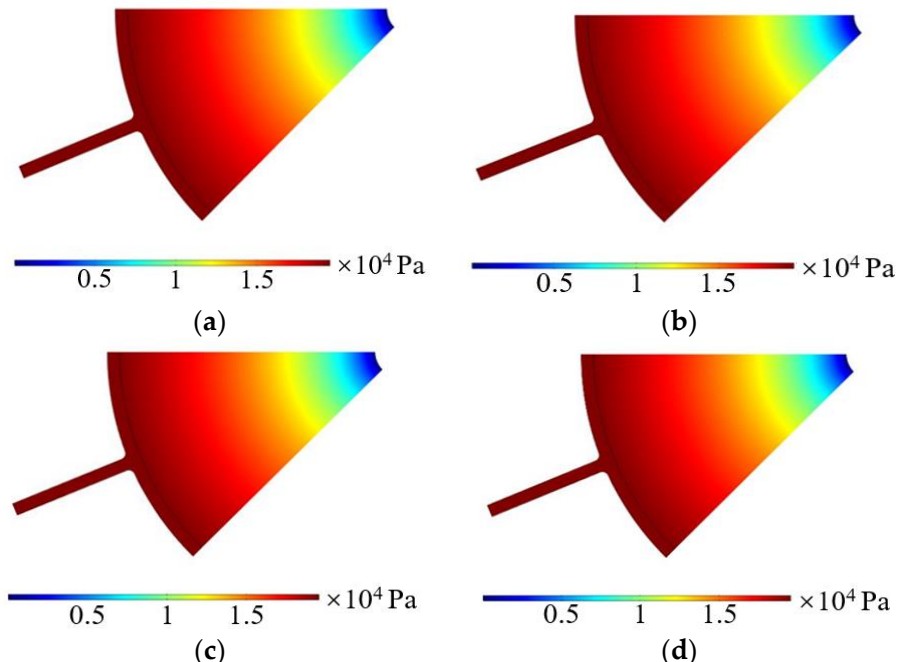

**Figure 18.** Pressure distribution at different electrode current densities. (**a**) Current density of 1200 A/m$^2$. (**b**) Current density of 1400 A/m$^2$. (**c**) Current density of 1600 A/m$^2$. (**d**) Current density of 1800 A/m$^2$.

Figure 20 displays the variation in the absolute value of the electrode overpotential. The overpotential ranges from 0.357 V to 0.491 V, demonstrating a gradual increasing trend. The absolute value of overpotential under optimal operating conditions is approximately 23% lower than the highest overpotential. This result is closely related to the increase in ohmic polarization caused by resistance. Figure 21 illustrates the change in potential with electrode current densities ranging from 1200 A/m$^2$ to 2000 A/m$^2$. It can be observed that the electrode with a current density of 1200 A/m$^2$ exhibits the highest potential of 2.066 V, which is 0.275 V higher than the potential of 1.791 V at 2000 A/m$^2$. The decrease in potential indicates an increase in voltage loss. This is because the ohmic polarization of resistance inevitably increases, resulting in a loss in cell voltage. Therefore, increasing the electrode current density is not conducive to improving battery performance. The appropriate value of current density should be selected based on the specific circumstances.

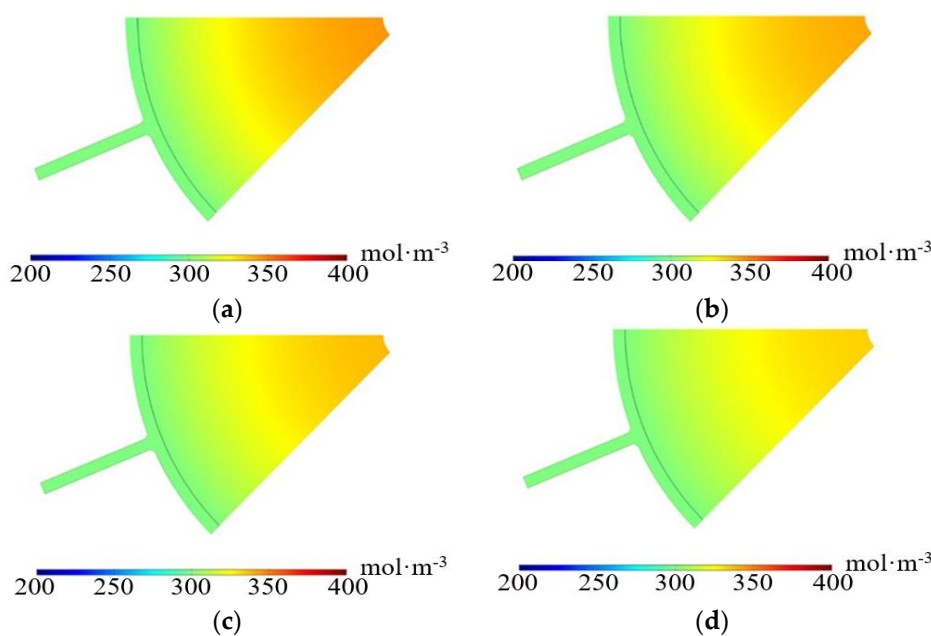

**Figure 19.** Concentration distribution of $V^{3+}$ at different electrode current densities. (**a**) Current density of 1200 A/m$^2$. (**b**) Current density of 1400 A/m$^2$. (**c**) Current density of 1600 A/m$^2$. (**d**) Current density of 1800 A/m$^2$.

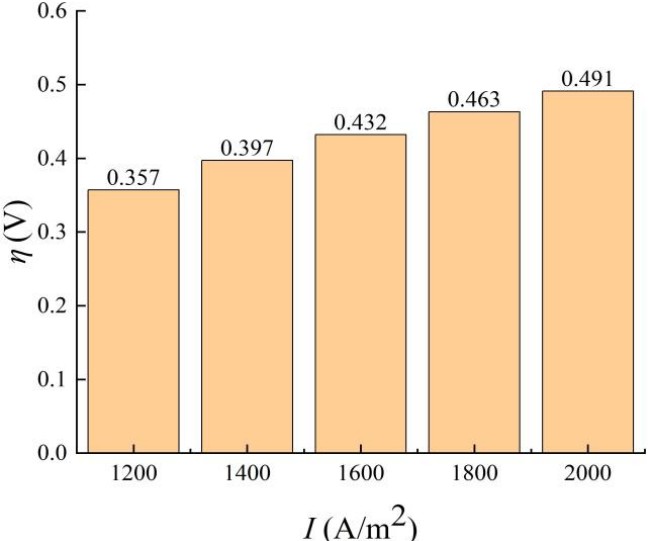

**Figure 20.** Variation in overpotential with electrode current density.

Figure 22 presents the variation in energy efficiency with changes in electrode current density. The calculations reveal that increasing the current density results in an increase in battery power loss. When the current density is 1200 A/m$^2$, the battery power loss is the lowest at 2.115 W. For other current densities, the battery power loss is as follows: 0.626 W; 0.667 W; 0.703 W; and 0.732 W. However, the increase in net power is too small to compensate for the power loss. Consequently, the energy efficiency decreases from 85.11% to 78.29%, indicating a downward trend. In terms of efficiency, it is more favorable to maintain the electrode current density at a lower level.

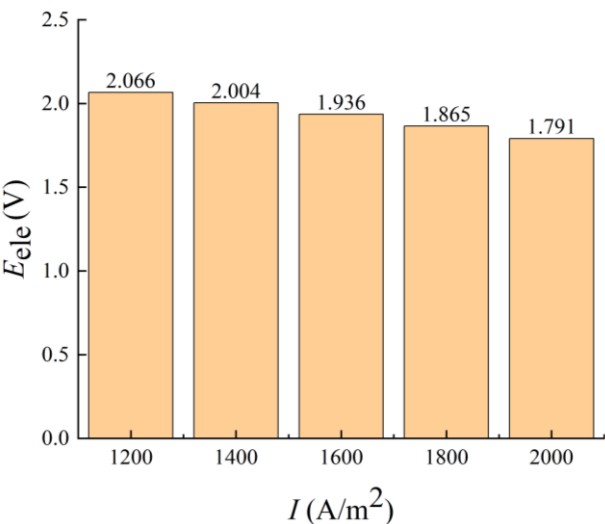

**Figure 21.** Variation in potential with electrode current density.

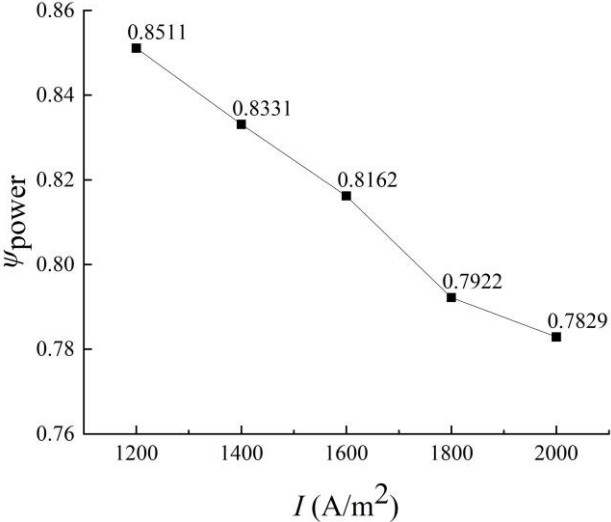

**Figure 22.** Variation in energy efficiency with electrode current density.

## 4. Conclusions

In the context of China's focused energy reform and power revolution, the development of safe, efficient, and reliable energy storage technologies has become crucial to achieving the goals of peak carbon and carbon neutrality. This paper utilized a numerical method to investigate the structural optimization and operational characteristics of VFB cells. The main conclusions of this study are as follows:

(1) The optimal electrode thickness for achieving the highest energy efficiency is determined to be 5 mm. The impact of changes in electrode thickness on battery performance is multifaceted. With an increase in electrode thickness, the electrode voltage drop and overpotential decrease while the potential increases. The overall energy efficiency exhibits a trend of initially increasing and then decreasing;

(2) Changes in imported electrolyte flow have various effects on battery performance. An increase in imported electrolyte flow results in a higher concentration of active ions, reduced concentration polarization, decreased overpotential, increased potential, increased pressure drop, increased pump loss, and decreased energy efficiency. Therefore, if higher potential and net power are desired, a larger imported electrolyte flow rate should be selected. Conversely, if higher energy efficiency is the priority, a smaller inlet electrolyte flow rate should be chosen;

(3) Higher imported electrolyte concentrations demonstrate improved battery performance. Increased electrolyte concentrations ensure an adequate supply of reactants, enhance material transport, and reduce concentration polarization. As the electrolyte concentration increases, overpotential and power losses decrease while potential and energy efficiency increase;

(4) The impact of changes in current density on battery performance is multifaceted. As the electrode current density increases, the concentration of trivalent vanadium ions increases. However, this also leads to an increase in overpotential, power loss, voltage loss, and a decrease in potential, net power, and energy efficiency. Therefore, it is advisable to choose a lower electrode current density when the electrochemical reaction process is operating optimally;

(5) The structural optimization and operative performance of the VFB cell are mainly conducted under design conditions with the univariate analysis method. For practical guidance, the effect of all the operative parameters on the electrochemical performance of the VFB cell should be analyzed via the multi-parameter optimization method.

**Author Contributions:** Conceptualization, L.C. and M.Q.; methodology, L.C., W.W., and L.Y.; software, K.S. and X.G.; validation, X.G., K.S., and W.W.; formal analysis, K.S., M.Q., and L.C.; investigation, K.S. and M.Q.; resources, L.C. and L.Y.; data curation, K.S. and Y.K.; writing—original draft preparation, K.S.; writing—review and editing, L.C. and L.Y.; visualization, Y.K.; supervision, X.D.; project administration, L.Y.; funding acquisition, L.C. All authors have read and agreed to the published version of the manuscript.

**Funding:** This research was funded by the National Natural Science Foundation of China, grant number 52006070.

**Institutional Review Board Statement:** Not applicable.

**Informed Consent Statement:** Not applicable.

**Data Availability Statement:** Not applicable.

**Conflicts of Interest:** The authors declare no conflict of interest.

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
