# Peer review of "Research on Performance Optimization of Novel Sector-Shape All-Vanadium Flow Battery"

_sustainability, doi:10.3390/su151914520_

Round 1

Reviewer 1 Report

In this manuscript, the authors introduce a novel sector-shaped all-vanadium flow battery with electrolyte distribution passages. They explore the influence of the electrode structure optimization on diverse physical fields and electrochemical performance through numerical simulations. However, there are some issues that need to be solved.

1.     To ensure a clear view of the velocity distribution and changes, please unify the colors in the velocity distribution bars for both Figure 2b and c.

2.     The accuracy of the numerical model in this manuscript has been validated by comparison with the radial design mentioned in reference 38, as shown in Figure 2c. What are the distinctive advantages of the model mentioned in this manuscript compared to the model mentioned in reference 38?

3.     The pressure distribution and concentration of V3+ distribution in figure 3 and 4 seems to be inconsistent with the description in lines 301-302 on page 8. Please determine the changes of electrode thickness in the model.

4.     The vertical axis in Figure 6 represents the cell potential, which appears to differ from electrode potential mentioned in line 340 on Page 10. Please specify.

Author Response

Dear Reviewer,

Thanks for your comments and suggestions on our manuscript. We have modified the manuscript accordingly and seriously. All changes have been marked by red in the revised manuscript. Detailed corrections are listed below point by point.

Best regards!

Reviewer 2 Report

The manuscript reported a mathematical and physical model that couples electrochemical reactions and thermal mass transfer processes within a novel sector-shape VRFB. Subsequently, the impact of cell thickness and operating parameters on the distribution of various physical fields and performance parameters has been investigated. It was found that the potential and overpotential decrease as the electrode thickness increases, while the energy efficiency initially rises and then decreases. As for operating parameters, higher electrolyte concentration demonstrates superior performance, while changes in electrolyte flow and current density have comprehensive effects on the battery. The cell performance can be adjusted based on the integrated mass transfer process and energy efficiency.

I consider the content of this manuscript will definitely meet the reading interests of the readers of the Sustainability journal. However, there are certain English spelling and grammar issues, and the discussion and explanation should be further improved. I suggest giving a minor revision and the authors need to clarify some issues or supply some more experimental data to enrich the content.

Detailed comments can be found in the PDf file. 

Author Response

(The authors gave the same response as above.)

Reviewer 3 Report

1. Figure 2. Unify the colors of the legend.

2. The maps are shown in a blurred version. Averaging would allow the outline places, e.g., with the highest factor concentration.

3. The authors often refer to the magnitude of the current flux and electric potential in the manuscript. Showing spatial maps of these quantities would significantly improve the clarity of the work.

4. rr. 81, "PR RG et al." The names of authors should not be abbreviations.

5. rr 85 "Additionally, the consumption of parasitic power resulted in significant losses." incomprehensible statement.

6. rr 117. "In this paper, a three-dimensional mathematical and physical model of the novel radial VFB cell was established based on our previous work [30]. The impact of cell thickness and operating parameters on the transport pro cess was analyzed. The results will facilitate a broader application of the VFBs in new energy systems."

Lack of clear indication of scientific goals and potential benefits. How the research differs from those mentioned in the introduction and what new they bring. Are the studies repeating them?

The introduction requires re-editing and sorting by loss causes and calculation methods used. So is the bibliography. Please see the "Quick Reference Formatting Guide." In addition, the computational model and mathematical apparatus used are very poor.

Minor editing of English language required

Author Response

(The authors gave the same response as above.)

Round 2

Reviewer 3 Report

The authors significantly improved the quality and clarity of the manuscript. The bibliography is correct. I have no comments on the manuscript.

Thanks.